# What Determines the District-Level Disparities in Immunization Coverage in India: Findings from Five Rounds of the National Family Health Survey

**DOI:** 10.3390/vaccines11040851

**Published:** 2023-04-16

**Authors:** Nandita Saikia, Krishna Kumar, Jayanta Kumar Bora, Souvik Mondal, Santosh Phad, Sumeet Agarwal

**Affiliations:** 1Public Health and Mortality Studies, International Institute for Population Sciences, Mumbai 400088, India; 2Centre for the Studies of Regional Development, Jawaharlal Nehru University, New Delhi 110067, India; 3VART Consulting Ltd. (P), Delhi NCR 201002, India; 4Department of Electrical Engineering, Indian Institute of Technology Delhi, New Delhi 110016, India

**Keywords:** immunization coverage, district level, Fairlie decomposition, socio-economic and healthcare utilization, inequalities, India

## Abstract

India’s Universal Immunization Programme has been performing at a sub-optimal level over the past decade, with there being a wide disparity in terms of immunization coverage between states. This study investigates the covariates that affect immunization rates and inequality in India at the individual and district levels. We used data from the five rounds of the National Family Health Survey (NFHS), conducted from 1992–1993 to 2019–2021. We used multilevel binary logistic regression analysis to examine the association between demographic, socio-economic and healthcare factors and a child’s full immunization status. Further, we used the Fairlie decomposition technique to understand the relative contribution of explanatory variables to a child’s full immunization status between districts with different immunization coverage levels. We found that 76% of children received full immunization in 2019–2021. Children from less wealthy families, urban backgrounds, Muslims, and those with illiterate mothers were found to have lower chances of receiving full immunization. There is no evidence that gender and caste disparities have an impact on immunization coverage in India. We found that having a child’s health card is the most significant contributor to reducing the disparities that exist regarding children’s full immunization between mid- and low-performing districts. Our study suggests that healthcare-related variables are more crucial than demographic and socio-economic variables when determining ways in which to improve immunization coverage in Indian districts.

## 1. Introduction

Immunization is the most essential and cost-effective medium through which to prevent diseases and deaths among children. It protects children against serious illness and vaccine-preventable diseases such as tuberculosis, diphtheria, pertussis, poliomyelitis, and measles [1]. The United Nations set childhood immunization coverage as a critical indicator of health in order to monitor the fourth Millennium Development Goal (MDG-4), which aimed to reduce the under-five mortality rate by two-thirds between 1990 and 2015. After that, in 2015, the United Nations proposed Sustainable Development Goal (SDG) 3, which aims to ensure good health and wellbeing for all, including universal immunization coverage [2]. The new goal also focuses on monitoring immunization coverage and inequalities associated with vaccination. Despite medical and technological advancement, children suffer from vaccine-preventable diseases due to disparities in vaccine coverage [3,4]. In addition, delayed vaccination has a severe impact on the disease burden [5]. A previous study documented that refusing or delaying vaccination contributes to disparities in vaccine uptake and coverage, which are both essential to controlling vaccine-preventable diseases. In 2019, the WHO listed “vaccine hesitancy” as one of the top ten threats to global health [6].

The use of vaccines has significantly lowered the child mortality rate worldwide. Studies have shown that the global under-five death rate has declined from 93 deaths per thousand live births in 1990 to 38 deaths in 2021 [7]. Globally, infectious diseases remain a leading cause of death among under five children [7], which can be prevented through timely vaccination. South Asian countries have shown significant progress towards reducing child mortality rates, which decreased from 129 per thousand live births to 53 per thousand live births between 1990 and 2015. However, two million under-five deaths were recorded in 2016, and half of these deaths were due to vaccine-preventable diseases [8,9]. Nearly one-tenth of children, about 12.9 million, did not receive any vaccine in 2016, and around 19.5 million infants missed the routine immunization services [10,11]. Moreover, dropping out vaccine courses poses a significant health risk to children; this is common in South Asian countries, and India is not an exception to these health challenges [9]. More under-five deaths occur in low and middle-income countries than anywhere else [12,13].

India’s government launched the Universal Immunization Programme in 1985 to protect infants and children from mortality caused by six preventable diseases [14]. Meanwhile, the Pulse Polio campaign was introduced in order to reduce the incidence of polio in India, drastically reducing the polio incidence by 2012. In 2014, the WHO applauded India’s efforts to eradicate polio and awarded the nation with polio-free status. In addition, intending to achieve 90% vaccine coverage by 2020, the government launched Mission *Indradhanush*, which aimed to vaccinate the underserved and children living in far-flung areas [15]. Despite the government’s efforts, full childhood immunization coverage is below a satisfactory level. The NFHS, a large-scale survey and a crucial source of information on immunization coverage, shows that full childhood immunization coverage has increased from 35% in 1993 to 76% in 2019–2021 [16,17]. However, there is a wide disparity in the coverage of different vaccination doses. In a previous study, socio-economic inequalities were significantly associated with child immunization in India [18].

Previous studies have primarily focused on the association between socio-economic and regional covariates, and immunization coverage. However, systematic investigations on immunization inequality are limited, and pathways leading to such imbalances are still unknown. We aim to investigate the significant covariates that are associated with a child’s full immunization status. Further, we examine the effects of individual and district-level characteristics on a child’s full immunization status and assess how they cause variation in immunization coverage at the district level in India. In addition, we aim to evaluate the relative impact of demographic, socio-economic and healthcare utilization variables on a child’s full immunization status between districts with different immunization coverage levels. This study may suggest adequate interventions for reducing immunization inequality and speed up efforts that aim to achieve immunization targets.

## 2. Material and Methods

### 2.1. Data Source

We used the dataset of five rounds of the National Family Health Survey (NFHS) conducted during 1992–1993 and 2019–2021. The NFHS is a cross-sectional survey that provides data on household populations and their socioeconomic characteristics at the individual level. In addition, data on the utilization of healthcare are available up to the district level. The survey was conducted by the Ministry of Health and Family Welfare, within the government of India. The International Institute for Population Sciences was the nodal agency for the surveys. Data on immunization were collected from the child’s health report card and from mothers who directly reported this information at each round of the NFHS. Detailed information on the sample design is available in the national report [17]. The survey collected information from a total of 636,699 Indian households from 29 states, 8 union territories (UT) and 707 districts, with a response rate of 98%. In the selected households, 724,115 women and 101,839 men were interviewed. We included a total sample of 43,291 children aged 12–23 months in the final analysis. We also used rural health statistics [19] for information regarding the Sub-Centre, Primary Health Centre (PHC) and Community Health Centre (CHC) at the district level. PHCs and CHCs are the basic units of the Indian Public Health Care System, serving 5000 people in plain areas and 3000 people in hilly areas; Sub-centres are the grass-root-level health care facilities in India. PHCs serve 30,000 people in plain areas and 20,000 people in hilly areas, and are usually equipped with a medical officer, 14 paramedical staff and 2 additional stuff nurses. A PHC is a referral unit for 5 to 6 sub-centers.

### 2.2. Outcome Variable

The outcome variable in this study was the full immunization status of children. Full immunization is defined as children aged 12–23 months who have received one dose of the BCG vaccine, one dose of the measles vaccine, three doses of the DPT (Diphtheria, Pertussis, Tetanus) vaccine, and the polio vaccine (excluding polio vaccine given at birth). We assigned a value of 1 for fully immunized children and 0 otherwise.

### 2.3. Predictors

Predictor variables were selected, as they have been used as significant covariates of child immunization in the previous literature. We considered demographic, socio-economic, and healthcare characteristics to identify the covariates associated with a child’s immunization status. Under the demographic variables, we included the sex of the child (male or female) and the birth order of the child (1, 2, 3, and 4 and above). We also included the mothers’ age at first birth (15–24, 25–34, 35–49 years). The household size was categorized as one–three family members, four–five family members, six–seven family members, and eight and above family members living together in a house. Further, the child’s place of residence (urban or rural) and region (north, central, east, north-east, west, and south) were also considered.

In socio-economic variables, we included the mother’s level of education (illiterate, primary, secondary, and higher), caste (ST, SC, OBC, and others), religion (Hindu, Muslim, and other), and wealth quintile (poorest, poorer, middle, richer and richest). We categorized media exposure into three categories (no, partial and full). No refers to respondents who were not exposed to at least one of the three forms of media (watching television, reading newspaper and listening to radio) once in a week. Partial refers to respondents who were exposed to any one or two of the three forms of media. Full refers to respondents who were exposed to all three forms of media.

Under healthcare utilization variables, we included the possession of a child’s health card (no or yes), whether the distance to a health facility was considered a problem by the respondent (no or yes) and the number of antenatal care visits (none, 1–3 and 4+). A health card is a document in which detailed information about a child, such as its date of birth, height, weight and immunization schedules, is available. The place of delivery refers to where the child was delivered and is categorized as a home, public or private facility. We also included mothers who had received tetanus toxoid injections (no or yes), those whose babies had received post-natal care after two months of delivery (no or yes), mothers who had received a post-natal checkup (categorized as a checkup performed within two days of delivery), and mothers who had received financial assistance after delivery (no or yes). Other healthcare variables included C-section delivery (no or yes) and the child’s size at birth (large, average, and small). C-section is the surgical delivery of a baby through an incision made in the mother’s abdomen and uterus.

Furthermore, we considered district-level factors, such as the number of sub-centers per thousand people in a district, the number of PHCs per thousand people in a neighborhood and the number of community health centers per thousand people in a district.

Moreover, we categorized India’s 707 districts into low-, medium-, and high-performing districts. Low-performing districts referred to districts in which the percentage of fully immunized children was below 50%. Medium-performing districts referred to districts in which the percentage of fully immunized children was 50% to 80%. High-performing districts referred to districts in which the percentage of fully immunized children was above 80%. Similarly, states were categorized as low-, medium-, and high-performing states regarding full childhood immunization status. The regions were categorized according to the six major geographical areas that have similar cultural settings. The six regions included the north (Jammu and Kashmir, Himachal Pradesh, Punjab, Haryana, Rajasthan, Delhi, Uttarakhand, and Chandigarh), the central area (Uttar Pradesh, Madhya Pradesh, and Chhattisgarh), the east (Bihar, West Bengal, Orissa, and Jharkhand), the northeast (Arunachal Pradesh, Assam, Manipur, Meghalaya, Mizoram, Nagaland, Sikkim, and Tripura), the west (Gujarat, Maharashtra, Goa, Dadra and Nagar Haveli and Daman and Diu) and the south (Andhra Pradesh, Karnataka, Kerala, Tamil Nadu, Andaman, and the Nicobar Islands, Lakshadweep, Puducherry, and Telangana).

### 2.4. Statistical Analysis

We detected a trend in the full immunization status of children in the last five rounds of the NFHS (NFHS-1 to NFHS-5). In addition, we ordered the states in terms of their performance regarding a child’s full immunization status for each round of the NFHS. Further, we presented a bivariate distribution in order to examine the association between demographic, socioeconomic and healthcare variables and a child’s full immunization status. A Pearson’s chi-square test was performed in order to identify the significant associations. We designed multilevel binary logistic regression models with random intercept and fixed slopes in order to calculate the adjusted odds ratio at three levels (individual, district, and state). When the *p*-value was 0.05, we considered the odds ratios to be significant at a 95% confidence interval.

We used demographic, socioeconomic and healthcare utilization variables at the individual level. In addition, we used three district level variables. A lower Akaike Information Criteria (AIC) value and a higher log-likelihood value were considered to determine the good fit of the model. We found that the mean VIF was equal to 1.40, indicating non-collinearity among the independent variables. The percentage of variance at the district and state levels was estimated using Intra Class Correlation (ICC). We analyzed the data using STATA 15. In addition, we used the lme4 package of R (version 4.0.2) for multilevel modelling. Further, we also mapped the district-wise proportion of fully immunized children using ArcGIS software(Version 10.5.0.6491, IIPS, Mumbai, India).

The mathematical formulation of the three-level model is shown below:$$logit\left(\pi_{ijk}\right)=\log\left(\pi_{ijk}/(1-\pi_{ijk}\;\right)=\beta_{0jk}+\beta_{1}x_{1ijk}+\beta_{2}x_{2ijk}+\cdots \cdots \beta_{n}x_{nijk}+u_{0jk}+v_{0jk}+e_{ijk}$$
where π*ijk* = p($Y_{ijk}=1$) is the probability of a child *i* in the district *j*, from state *k*, having a registered birth.

$Y_{ijk}$ would equal one if a child was registered, otherwise 0. The probability was defined as a function of an intercept and the explanatory variables. $\beta_{0jk}=\beta_{0}+\mu_{0jk},\;$where $\beta_{0jk}$ shows that the intercept was random at the $j^{th}$ (district) and $k^{th}$ (state) levels. The variables $x_{1ijk}$ to $x_{nijk}$ were exploratory variables, and their corresponding regression coefficients ($\beta_{1},\beta_{2},\ldots .\beta_{n})\;$were fixed effects.

$u_{0jk}$ is the random state effect assumed to be normally distributed with N (0, σu^2^)

$v_{ojk}$ is the random district effect assumed to be normally distributed with N (0, σv^2^)

$e_{ijk}$ is the random errors assumed to be normal with N (0,  σe^2^), independent of random effects at level 2 and level 3.

We further used Fairlie decomposition [20] analysis to understand the relative contribution of the predictor variables to a child’s full immunization status. The Fairlie decomposition technique is considered appropriate for binary models when aiming to decompose the contribution of each factor to immunization coverage. For the decomposition analysis, we combined the districts based on the immunization coverage level and referred to them as “high-performing district”, “medium performing district” and “low-performing district”. We designed three Fairlie decomposition models (medium-low, high-medium and high-low) in this study. A positive sign in the coefficient of the predictors indicated that the particular variable considered contributed to widening the disparity in the full immunization status of children between medium and low-performing districts, and vice versa. The details of the decomposition are given in Appendix A. In addition, we presented the mean value of the demographic, socioeconomic, and healthcare utilization variables by high-, medium, and low-performing districts in Appendix A.

## 3. Results

### 3.1. Levels and Trends of Child’s Immunization

Figure 1 shows that the full immunization coverage of children has consistently increased over time, from 35% in 1992–1993 (NFHS 1) to 76% in 2019–2021 (NFHS-5). The full immunization coverage disparity between male and female children showed a considerable reduction from NFHS-1 to NFHS-5. 

Table 1 shows that the number of states with a lower full immunization coverage was nearly identical during the first three rounds of the NFHS. In NFHS-5, the number of states with a lower full childhood immunization coverage was substantially reduced (from fourteen states in NFHS-1 to zero states in NFHS-5). In contrast, the number of states with a medium level of full immunization coverage increased from 11 states in NFHS-1 to 22 states in NFHS-5. Similarly, since the first round of the NFHS, the number of states with a higher level of childhood immunization coverage increased (no states with a higher coverage of full immunization in NFHS-1 compared to 14 states in NFHS-5). Among the 43,291 children aged 12–23 months, 76% of children were fully immunized in NFHS-5 (2019–2021).

Figure 2 shows that out of the 707 districts of India, 50 districts showed a full immunization status that was lower than 50%. We observed three distinct geographical clusters present across the boundaries of various states. We observed the first cluster in the north-western part of the country, covering the districts of Rajasthan, Gujarat and some parts of Madhya Pradesh and Maharashtra. The second cluster with a poorer performance was observed in the north-eastern part of India, including Arunachal Pradesh and parts of Assam. In addition, Madhya Pradesh showed a poor immunization coverage, with 30 out of 51 of its districts recording an immunization rate lower than 80%. Interestingly, a medium level of immunization status (lower than 80%) was also found in demographically advanced states such as Tamil Nadu and Karnataka.

### 3.2. Determinants of Immunization in India

Table 2 shows that over three-fourths (76%) of children received full immunization in 2019–2021. We found a marginal sex difference regarding full childhood immunization coverage. However, as the child’s birth order rises, the coverage gradually decreases (from 79% of children first in birth order compared to 68% of children fourth and above in birth order). A higher proportion of children (83%) born to mothers aged 35–49 years were found to be fully vaccinated compared to children (74%) born to mothers aged 15 –24 years. The family size was significantly negatively associated with the child’s full immunization status. The percentage of fully immunized children was found to be lower in the north–east region (66%) and eastern region (72%). On the other hand, the full immunization rate was higher than the national level in the west (75%), south (78%), north (79%) and central (82%) regions.

Furthermore, socio-economic variables were significantly associated with a child’s full immunization status. Nearly 68% of children whose mothers did not receive formal schooling had full immunization status, whereas 78% of children whose mothers had a higher level of education had full immunization status. We did not find that caste had a considerable impact on a child’s immunization status. Full vaccination status among Muslim children was lower (71%) than Hindus (77%). Mothers’ media exposure positively influenced the child’s full immunization status. In addition, we found that children belonging to the richest wealth quintile showed a higher immunization rate (78%) than children of the poorest wealth quintile (71%).

Moreover, healthcare utilization variables were significantly associated with a child’s full immunization status. Children with a health card were more likely to be vaccinated (80%). In addition, looking at the distance between a child’s home and a health facility, more than three-fourths (77%) of children were found fully vaccinated when mothers reported that their residence was near a health facility. A high proportion of children were fully vaccinated when their mothers had received at least four antenatal care visits, tetanus toxoid (TT) injections, when they had been delivered in a public health facility and when they had received a post-natal check within two days of delivery. In addition, we found that children were more likely to be fully immunized if they had received post-natal care after birth than if they had not (81% vs. 72%).

Interestingly, the child’s full immunization status was higher (81%) among mothers who had received financial assistance after delivery. We found a significant disparity in full immunization coverage between children born via C-section (79%) and children born via conventional delivery (75%). A child’s full immunization rate was also marginally higher among children who had an average size at birth (75%) than in children who had a small size at birth (73%).

We found a random variance of 0.16 (Standard Error 0.41), 0.17 (0.40), and 3.29 (1.81) at the state, district, and individual levels, respectively (Table 3). The ICC at the state level was 0.04 (0.20), showing that 4% of the total variation in a child’s full immunization status was explained by state-level differences, and the remaining 96% variation lay within states. Furthermore, 5% of the total variation in a child’s full immunization status was due to differences between districts.

Among demographic variables, we found that a child’s sex and birth order were not significantly associated with his or her full immunization status. Interestingly, the odds of a child receiving full immunization were higher among rural residents. Mothers aged 20–34 years at their first birth were more likely to have a fully immunized child compared to mothers aged 15–19 years. We found that, compared with children belonging to households with a size of 1 to 3, children belonging to households with a size of eight and above were less likely to have received full immunization.

Compared with mothers who had received no formal education, mothers who had primary, secondary and higher education were more likely to have a child with full immunization status. Looking at religion, we found that, compared to Hindus, children who belonged to Muslim families were less likely to have full immunization status. We found that the wealth quintile was significantly associated with a child’s full immunization status.

Furthermore, we found that healthcare utilization factors were significantly correlated with a child’s full immunization status. Children who had a health card were more likely to have full immunization status. Mothers who had received tetanus toxoid (TT) injection, delivered their child in a health facilities and had received a post-natal checkup were more likely to have a child with full immunization status. If a child had received a baby postnatal checkup after two months, he or she was significantly more likely to have full immunization status.

There was only one variable at the district level that showed statistically significant results. We found that children living in a district in which a higher number of community health centers (CHC) per thousand people were available were more likely to have full immunization status.

### 3.3. Decomposition Results

Table 4 shows that the likelihood of a child having full immunization was higher among medium performing districts than low-performing districts. Model 1 shows that the included variables explained 35% of the full immunization disparity between medium and low-performing districts.

We found that the possession of a child’s health card (69%) was the most significant contributor to widening the disparity in children’s full immunization status between medium and low-performing districts, followed by mothers who had received TT (10%), baby postnatal checkup after two months (7%) and a mother’s postnatal checkup (3%). On the other hand, household size (2%) and religion (3%) contributed to reducing the disparity in children’s full immunization status.

Districts with a higher number of CHCs per thousand people reduced the disparity in children’s full immunization status (0.4%). Additionally, we considered other healthcare utilization variables at district levels, such as the number of sub-health centers and primary health centers per thousand people. However, the contribution of these variables was not significant. We found that healthcare-related variables were of the highest importance in explaining the immunization disparity.

Model 2 shows that the included variables explained 26% of the full immunization disparity between medium and high-performing districts. Similar to model 1, in model 2, 85% of the disparities in the immunization status between medium and high-performing districts were due to healthcare-related variables.

Furthermore, model 3 showed that the predictor variables explained 28% of the full immunization disparity between the high- and low-performing districts. Some demographic variables, such as the place of residence (7%), widened the disparity, and household size (2%) reduced the disparity in children’s immunization status. Out of the explained disparities, healthcare variables explained about 85% of the immunization disparities between high- and low--performing districts.

## 4. Discussion

This is the first study using the Fairlie decomposition technique to show the relative contribution of demographic, socio-economic and healthcare utilization variables to a child’s full immunization status between districts with different immunization coverage levels. We found that full childhood immunization coverage improved from 35% percent in 1992–1993 to 76% in 2019–2021. Our study showed that children’s full immunization coverage was below 50% in 50 districts out of the 707 districts of India. Many districts, including Madhya Pradesh, Uttar Pradesh, Rajasthan, Gujarat, and the North-East states, showed a poorer performance in terms of their full childhood immunization coverage. The ICC at the district level was 5%, indicating the need to initiate an adequate immunization program at the district level. Previous studies have also shown lower immunization coverage than India’s national average in BIMARU states (Bihar, Madhya Pradesh, Rajasthan, and Uttar Pradesh) [21]. Surprisingly, we found a lower level of immunization in demographically advanced states such as Tamil Nadu and Karnataka.

Interestingly, the multilevel model result showed that there was no significant difference in the immunization rate by sex. There has been a notable change in recent years, as previous studies documented immunization rates that were higher among male children [22,23,24]. Such a shift in the immunization status among female children may be due to improved coverage and the uplift in the social status of female children in India in recent years. We suggest that a further qualitative study on the states in which gender inequality is evident. The multilevel model of our study did not show a significant association between a child’s birth order and his or her full immunization. In contrast, a previous study showed lower immunization rates among children with a higher birth order [25,26].

Unlike the previous study, this study showed that children of rural areas were more likely to have received full immunization. Often, rural people do not demand that their children receive the recommended vaccines because of inadequate knowledge regarding the required vaccines [27]. A higher immunization coverage in rural areas could be due to the strengthening of PHCs in rural areas and the offering of financial incentives to mothers who delivered their child in public health facilities. Janani Suraksha Yojana (cash transfer during pregnancy) has increased the proportion of women who give birth in public health facilities [28]. As expected, a higher number of deliveries in facilities was significantly positively associated with higher immunization coverage [29]. Mothers who had received education at the primary level and above were significantly more likely to have a child with full immunization status. Higher levels of education among mothers led to increased access to healthcare services and consequently, adequate information on the required vaccinations for children. A previous study showed that educated mothers had better knowledge about the availability and the necessity of different kinds of health services than their counterparts [1,25].

This study showed that children in Muslim families were likely have full immunization status. People’s particular customs and traditional beliefs may have also led to vaccine hesitancy [30,31]. Religious beliefs have been significantly associated with immunization inequality in low-, middle- and high-income countries [32]. A recent study revealed that religious affiliation and the likelihood of receiving a certain vaccine were significantly correlated in Germany [33]. In addition, deprived groups, such as ST people in India, were more likely to live in remote areas with limited immunization facilities. However, we did not find a significant correlation between a child’s caste and his or her full immunization status. The expansion of the social mobilization network approach that was used in the National Polio Programme, which involved famous political leaders, sportspeople, and actors, could improve trust between parents and healthcare providers, as suggested in previous studies [34]. The household’s wealth status contributed to determining whether child had full immunization status. Previous studies have shown that belonging to a more privileged class improves the accessibility of health centers and thereby increases immunization rates among children [35,36]. Children that had a health card were more likely to be vaccinated. A health card is a useful health record that contains information about a child’s height and weight by age at monthly intervals. Usually, it gives scope to compare the growth of the indexed child to the average growth of children at the same age. Further, it contains information about vaccination dates and dosage. Both the parents and vaccination providers may benefit from such cards regarding the vaccination of children. This is because it records the date of all the child’s completed vaccines and hence keeps track of all the child’s upcoming vaccinations. Parents that do not receive the health card or that lose the health card may be deprived of timely information regarding their child’s vaccination. The possession of a child’s health card can be considered a proxy in order to easily access the healthcare system in general. In line with our study, a previous study showed that the likelihood of a child receiving vaccination is correlated with the family’s distance from a health facility in Nigeria [37].

In addition, the decomposition result of this study highlighted that among the individual or household variables, the mother’s education, religion and wealth played a relatively important role in explaining the immunization disparity between two groups of districts. However, the healthcare-related variables were of the highest importance in explaining such disparities. For example, about 85% of the disparities in full childhood immunization between high- and low-performing districts were due to healthcare-related variables. This indicates that healthcare-related variables are more crucial than demographic and socio-economic variables when aiming to improve immunization coverage. Thus, our study’s findings suggest that improving health facilities and their accessibility is essential in order to improve immunization coverage in low-performing districts.

Although this study exhibited some crucial findings for policy making, there are some limitations. First, we used cross-sectional data, which prohibited the performance of causal inference using the experimental observations. Second, the childhood immunization information used was based on the vaccination report card and maternal recall. Previous studies have shown that maternal records tend to overestimate and vaccination report cards tends to underestimate the immunization coverage [38,39]. In addition, a previous study showed that indirect costs, such as traveling costs or long waiting times at vaccination centers, are barriers to fulfilling immunization coverage [24]. However, this study did not consider such factors because there are no data on waiting times and the health system performance.

Further research that assesses the supply-side and demand-side barriers to immunization and investigates the casual factors is required in order to inform decision makers regarding the uptake of routine immunization, particularly in low-performing districts.

## 5. Conclusions

We found that there is no evidence of gender and caste disparity having an impact on immunization coverage. However, other important demographic and socio-economic variables, such as place of residence, mothers’ education, household size, household wealth status and religion, were significantly associated with the immunization status. In addition, the mother’s level of health care utilization and possession of a health card increased the likelihood of her child having full immunization status. The findings of the present study emphasize that children’s socio-economic condition can be improved via the mothers’ education and household wealth status. At the same time, we strongly suggest that ensuring the provision of a child’s health card could improve the full immunization coverage rate. Providing financial assistance to mothers after a child’s birth could also increase the coverage rate of full immunization.

## Figures and Tables

**Figure 1 vaccines-11-00851-f001:**
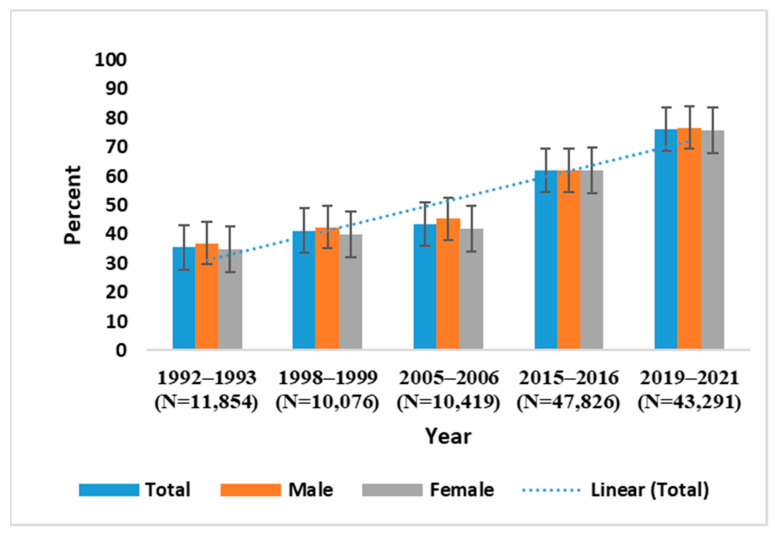
Full immunization status among children by sex aged 12–23 months by NFHS rounds.

**Figure 2 vaccines-11-00851-f002:**
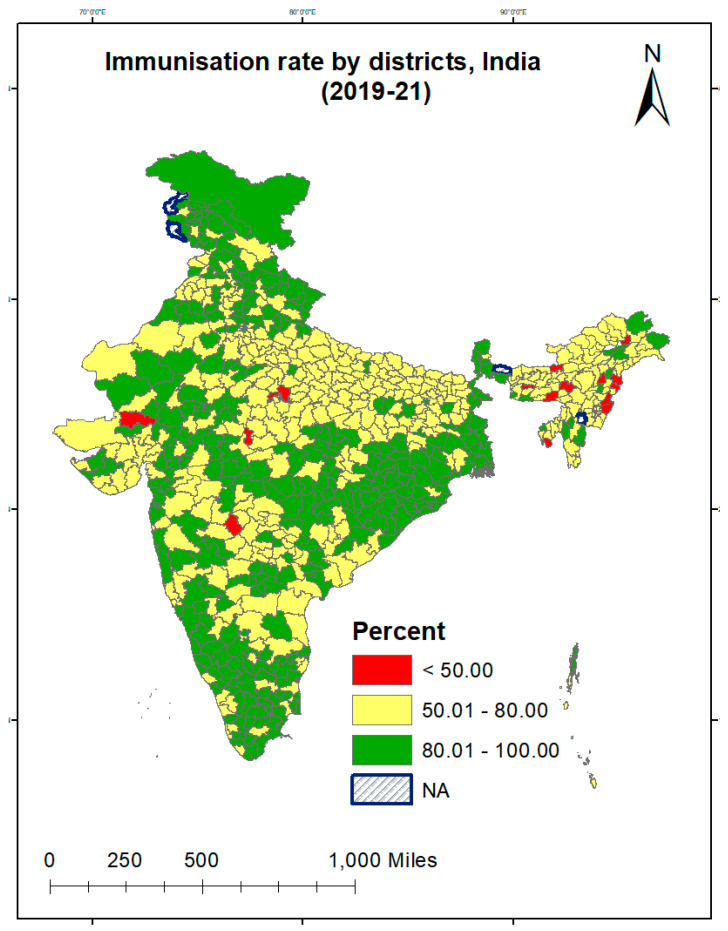
Percentage of children aged 12–23 months with full immunization status by district, 2019–2021, India. Data Source: Author created this map using GISArc software.

**Table 1 vaccines-11-00851-t001:** Number of states with full immunization status by NFHS rounds.

	Number of States *				
Immunization Status	NFHS 1	NFHS 2	NFHS 3	NFHS 4	NFHS 5
Low	14 (56.0%)	15 (58.0%)	15 (52.0%)	4 (11.0%)	0 (0.0%)
Medium	11 (44.0%)	9 (35.0%)	13 (45.0%)	25 (69.0%)	22 (61.1%)
High	0 (0.0%)	2 (7.0%)	1 (3.4%)	7 (19.4%)	14 (38.8%)
Total	25	26	29	36	36

* We presented the number of states as mentioned in NFHS. We have not made any adjustment regarding the number of states in each round.

**Table 2 vaccines-11-00851-t002:** Percentage of children aged 12–23 months who received full immunization by baseline characteristics, NFHS 2019–2021, India.

Independent Variables	Percent	Frequency	Chi-2 Value	*p*-Value
Demographic Variables				
Sex of the child			5.47	0.019
Male	76.6	22,495		
Female	75.7	20,796		
Birth order			296.50	0.000
1	78.9	17,194		
2	76.8	14,825		
3	73.1	6461		
4 and above	68.3	4810		
Place of residence			0.20	0.649
Urban	75.2	11,697		
Rural	76.5	31,595		
Mother’s age at first birth			89.91	0.000
15–24	74.4	14,035		
25–34	77.0	29,035		
35–49	82.8	220		
Household Size			92.53	0.000
One–three	78.4	3998		
Four–five	78.1	15,683		
Six–seven	76.5	12,593		
Eight and above	72.3	11,018		
Region			601.05	0.000
North	79.3	5654		
North East	65.9	1535		
Central	81.7	7289		
East	71.7	11,989		
South	77.9	11,505		
West	74.5	5319		
Socio-economic variables				
Mother’s education level			424.48	0.000
Illiterate	67.9	8160		
Primary	75.0	4887		
Secondary	78.7	22,731		
Higher	78.3	7513		
Caste			52.16	0.000
ST	76.2	10,089		
SC	76.2	4324		
OBC	76.7	18,642		
Others	75.1	10,236		
Religion			264.52	0.000
Hindus	77.1	34,435		
Muslims	71.0	6975		
Others	78.6	1881		
Mother’s media exposure			352.72	0.000
No	72.7	21,357		
Partial	79.6	21,335		
Full	75.3	599		
Wealth Quintile			424.50	0.000
Poorest	70.8	10,355		
Poorer	74.9	9226		
Middle	79.4	8560		
Richer	79.2	8141		
Richest	78.1	7009		
Healthcare variables				
Child’s health card			5200.00	0.000
No	13.7	2173		
Yes	79.5	41,118		
Distance to health facility problem			47.53	0.000
No	76.6	32,575		
Yes	74.9	10,716		
Number of antenatal care visit			1000.00	0.000
None	60.3	2394		
One-three	71.9	14,243		
Four and above	80.6	24,238		
Place of delivery			718.50	0.000
Home	64.0	4030		
Public facility	78.1	27,262		
Private facility	75.9	11,997		
Mother received Tetanus Toxoid injections			398.46	0.000
No	63.2	1829		
Yes	77.0	38,860		
Baby post-natal checkup within 2 months			407.08	0.000
No	72.4	21,640		
Yes	80.8	19,222		
Mother post-natal check			9.68	0.002
Within 2 days	77.1	6827		
After 2 days	76.0	36,464		
Received financial assistance			162.00	0.000
No	75.3	21,225		
Yes	80.5	16,403		
C-section delivery			99.31	0.000
No	75.2	33,116		
Yes	79.4	10,174		
Size of child at birth			76.12	0.000
Large	75.2	8328		
Average	76.9	29,997		
Small	73.1	4965		
Total	76.2	43,291		

Note: Total frequency of some independent variables (Antenatal visit, TT and baby postnatal care) are not equal to 43,291 due to missing cases.

**Table 3 vaccines-11-00851-t003:** Multilevel logistic binary regression results of child’s full immunization status by demographic, socio-economic, and healthcare variables, 2019–2021, India.

	Full Model			
Fixed Effects	AOR	Lower	Upper	*p*-Value
Independent Variables				
Intercept	0.02	0.01	0.03	0.000
Demographic and socio-economic variables				
Sex				
Male	Reference			
Female	0.98	0.93	1.03	0.390
Birth order				
1	Reference			
2	0.97	0.91	1.03	0.290
3	0.98	0.91	1.07	0.720
3+	1.01	0.92	1.11	0.840
Place of residence				
Urban	Reference			
Rural	1.28	1.19	1.38	0.000
Mother’s age at first birth				
15–19	Reference			
20–34	1.09	1.03	1.15	0.000
35–45	1.08	0.76	1.53	0.660
Mother’s education				
Illiterate	Reference			
Primary	1.14	1.04	1.25	0.010
secondary	1.22	1.13	1.32	0.000
Higher	1.18	1.06	1.31	0.000
Household size				
One–three	Reference			
Four–five	1.00	0.91	1.11	0.950
Six–Seven	0.96	0.87	1.06	0.430
Eight and above	0.79	0.72	0.88	0.000
Religion				
Hindus	Reference			
Muslims	0.74	0.68	0.81	0.000
Others	0.95	0.83	1.09	0.460
Caste				
SC	Reference			
ST	0.97	0.88	1.08	0.580
OBC	1.06	0.98	1.14	0.130
Others	0.99	0.90	1.08	0.750
Media exposure				
No	Reference			
Partial	1.10	1.03	1.17	0.000
Full	0.87	0.69	1.10	0.240
Wealth Quintile				
Poorest	Reference			
Poorer	1.11	1.03	1.20	0.000
Middle	1.30	1.18	1.42	0.000
Richer	1.40	1.26	1.55	0.000
Richest	1.44	1.27	1.64	0.000
Health Care Variables				
Mother received TT injection				
No	Reference			
Yes	1.48	1.33	1.65	0.000
Mother received ANC				
No	Reference			
One–three	1.25	1.13	1.39	0.000
Four and above	1.60	1.44	1.79	0.000
Place of delivery				
Home	Reference			
Public	1.38	1.27	1.50	0.000
Private	1.16	1.05	1.28	0.000
Mother’s postnatal checkup				
Within 2 days	Reference			
After 2 days	1.16	1.07	1.26	0.000
Baby postnatal checkup within 2 months				
No	Reference			
Yes	1.38	1.29	1.47	0.000
Have a health card				
No	Reference			
Yes	26.73	23.08	30.96	0.000
Distance to heath facility				
Not a problem	Reference			
Big Problem	0.98	0.92	1.04	0.520
District level variables				
Sub-centres per thousand	1.00	1.00	1.00	0.060
PHC per thousand	1.00	1.00	1.00	0.730
CHC per thousand	0.98	0.97	1.00	0.020
Random effect				
State level	0.16	0.41		
District level	0.17	0.40		
Individual level	3.29	1.81		
ICC				
State level	0.04	0.20		
District level	0.05	0.22		
VIF	1.40			
AIC	38,360.60			
Log-likelihood	−19,141.30			
District	707			
State	36			
Total observation	40,892			

**Table 4 vaccines-11-00851-t004:** Decomposition analysis of child’s immunization status by demographic, socio-economic and healthcare factors associated with child’s full immunization status, 2019–2021, India.

Independent Variables	Medium vs. Low-Performing Districts (Model 1)		High- vs. MediumPerforming Districts (Model 2)		High vs. Low-Performing Districts (Model 3)	
	Contribution Coefficient	Contribution Coefficient in Percentages	Contribution Coefficient	Contribution Coefficient in Percentages	Contribution Coefficient	Contribution Coefficient in Percentages
Demographic variables						
Sex	−0.036	−0.8	0.050	0.9	0.050	1.0
Birth order	0.001	0.0	−0.005	−0.1	−0.005	−0.1
Place of residence	0.275 ***	6.3	0.364 ***	7.1	0.364 ***	7.1
Mother’s age at first birth	0.047	1.1	0.063	1.2	0.063	1.2
Household size	−0.089 ***	−2.0	−0.108 ***	−2.1	−0.108 ***	−2.1
Total contributions by demographic variables		4.2		5.0		5.0
Socio-economic variables						
Mother’s education	0.082 ***	1.9	0.058	1.1	0.058 *	1.1
Caste	−0.028	−0.6	−0.019	−0.4	−0.019	−0.4
Religion	−0.125 ***	−2.8	−0.001	−0.0	−0.001	−0.0
Media Exposure	0.072 **	1.6	0.121	2.3	0.121 **	2.3
Wealth	0.069 ***	1.6	0.076 ***	1.5	0.076 ***	1.5
Total contributions by socio-economic variables		0.6		1.5		4.9
Healthcare variables						
Has a health card	3.014 ***	68.6	3.604 ***	69.9	3.604 ***	70.0
Distance to health facility	0.011	0.2	−0.042	−0.8	−0.042	−0.8
Antenatal care	0.238	5.4	0.228 ***	4.4	0.228 ***	4.4
Place of delivery	−0.002	−0.1	0.045	0.9	0.045	0.9
Tetanus toxoid Injections	0.435 ***	9.9	0.203	3.9	0.203	3.9
Baby post-natal care	0.300 ***	6.8	0.310 ***	6.0	0.310 ***	6.0
Mother’s post-natal care	0.148 ***	3.3	0.218 ***	4.2	0.218 **	4.2
Sub-centre per thousand people in a district	0.000	0.0	0.000	0.0	0.000	0.0
Primary health centre per thousand people in a district	0.000	0.0	0.001	0.0	0.001	0.0
Community health centre per thousand people in a district	−0.016 ***	−0.3	−0.012	−0.2	−0.012	−0.2
Total contributions by healthcare variables		94		85		85
Summary of Fairlie decomposition						
Mean predictor of lower immunized district	0.69		0.88		0.88	
Mean predictor of medium immunized district	0.46		0.69		0.46	
Row difference	0.23		0.19		0.42	
Total explained	0.08		0.05		0.12	
Percent of explained disparity in immunization between lower immunized district and medium immunized district	34.5		25.6		27.5	
Percent of unexplained disparity in immunization between lower immunized district and medium immunized district	65.4		74.3		72.4	
Number of observations	24,479		40,149		17,156	

Note: *** *p* < 0.01, ** *p* < 0.05, * *p* < 0.10.

## Data Availability

Authors analyzed the secondary data accessible from the DHS website with a request to use them for research purposes only. https://dhsprogram.com/data/ (accessed on 10 December 2022).

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
