# Peer review of "What Determines the District-Level Disparities in Immunization Coverage in India: Findings from Five Rounds of the National Family Health Survey"

_vaccines, 2023, doi:10.3390/vaccines11040851_

Round 1

Reviewer 1 Report

In this exhaustive study, the authors investigate various demographic and socio-economic covariates that affect immunization rates at district levels. The study is well designed and executed with the interesting data.

The authors suggest that having a child’s health card is the significant factor in improving the immunization rate but they did not discuss how and why that is the case. The authors need to elaborate on this. 

Author Response

Thank you for your positive comments. In the revised draft, we have included a paragraph in lines 386-394 explaining how a health card can help with vaccination.

Reviewer 2 Report

This paper evaluates community-level vaccination coverage in India.

The results are easy to understand and there are no major problems with the textual description, but there is one problem with the concept of the analysis.

The analysis is very detailed and deals with a large number of factors in the analysis. Each of the factors seems to be related and confounding, but no consideration is given to this point. It would be better to carefully describe the analysis.

Author Response

We have carefully edited the results section again. There are no confounding factors since we did a multi-collinearity test by calculating VIF (mean VIF=1.40).

Reviewer 3 Report

The article reviews the evolution of vaccination in India and the determinants associated with a complete vaccination of young children, beyond the usual characteristics.

The article is well written, clear, interesting but too long. It would gain to be more synthetic. Everything that is not essential should be deleted, especially the repetition of all the data that are in the figures or tables, the introductory sentences to these figures or tables...(see my suggestions in the text).

Few typing erros to correct. Some paragraphs are not in the correct chapter The conjugation should be standardized: use the past or present tense. A sentence in the conclusion is not appropriate and the conclusion from the summary and the article are sligthly different : I would emphase the conclusion of the summary !

Author Response

We have revised the manuscript in light of the comments here.